# The Role of STING in Liver Injury Is Both Stimulus- and Time-Dependent

**DOI:** 10.3390/nu14194029

**Published:** 2022-09-28

**Authors:** Kevin Siao, Dounia Le Guillou, Jacquelyn J. Maher, Caroline C. Duwaerts

**Affiliations:** 1Department of Medicine, University of California, San Francisco, CA 94143, USA; 2The Liver Center, University of California, San Francisco, CA 94143, USA

**Keywords:** NASH, STING, FPC diet, insulin resistance, toxic liver injury

## Abstract

STING, *Tmem173*, is involved in liver injury caused by both infectious and sterile inflammatory models. Its role in toxic liver injury and non-alcoholic fatty liver disease (NAFLD), however, is less clear. While a few groups have investigated its role in NAFLD pathogenesis, results have been conflicting. The objective of this study was to clarify the exact role of STING in toxic liver injury and NAFLD models. Goldenticket mice (*Tmem173^gt^*), which lack STING protein, were subjected to either a toxic liver injury with tunicamycin (TM) or one of two dietary models of non-alcoholic fatty liver disease: high fructose feeding or Fructose-Palmitate-Cholesterol (FPC) feeding. Three days after TM injection, *Tmem173^gt^* mice demonstrated less liver injury (average ALT of 54 ± 5 IU/L) than control mice (average ALT 108 ± 24 IU/L). In contrast, no significant differences in liver injury were seen between WT and *Tmem173^gt^* mice fed either high fructose or FPC. *Tmem173^gt^* mice only distinguished themselves from WT mice in their increased insulin resistance. In conclusion, while STING appears to play a role in toxic liver injury mediated by TM, it plays little to no role in two dietary models of NAFLD. The exact role of STING appears to be stimulus-dependent.

## 1. Introduction

Stimulator of interferon genes (STING, *Tmem173*), a classical nucleic acid sensor of DNA, is located on the membrane of the endoplasmic reticulum. Its C-terminal tail is located in the cell cytosol and allows STING to translocate to the Golgi when activated. STING has been linked to both exogenous pathogens, such as DNA viruses, but also endogenous DNA, such as mtDNA, sensing in the body [1]. In the liver, specifically, STING has been linked to both sterile and non-sterile inflammation, as caused by either ethanol or hepatitis virus, respectively. STING has a complex downstream molecular pathway that is dependent on the activating stimuli. The simplest pathway includes STING activation and oligomerization, phosphorylation of TBK1 (tank binding kinase 1), subsequent phosphorylation of IRF3 (IFN-regulatory factor 3), translocation of IRF3 to the nucleus, and transcription of numerous interferon type-I genes such as interferon-β (*Ifnb*). However, STING can also lead to cell death. This cell death signaling cascade also begins with the phosphorylation of TBK1 but leads to the uncovering of a BH3-only domain on IRF3, which permits the interaction of IRF3 with BAX/BAK (BCL2 associated X, apoptosis regulator and BCL2 antagonist/Killer 1) on the mitochondria, leading to mitochondrial permeability transition pore opening and activation of the mitochondrial-dependent cell death cascade [2,3,4].

Recently, STING has been connected to hepatic sterile inflammation by several groups. For instance, STING has been shown to worsen liver outcome following toxic liver injury such as induced by ethanol or CCl_4_ [5,6,7,8]. The role of STING during metabolic liver injury as induced by high fat diets has also been briefly studied, but results have been inconsistent [9,10,11]. Certain studies have demonstrated a detrimental role for STING during high fat diet (HFD) feeding or methionine- and choline-deficient (MCD) diet feeding [10,11], while others have demonstrated a potential benefit to STING during HFD feeding [9].

We thus set out to determine the exact role that STING plays during different liver injury models. Notably, we administered a toxic liver injury model—tunicamycin (TM), which leads to acute injury. We then decided to further pursue the role of STING in dietary-induced NASH models: high fructose diet and the Fructose–Palmitate–Cholesterol (FPC) diet. We chose these models as TM mimics a toxic liver injury while the high fructose and FPC diets mimic NAFLD. In order to study the role of STING, Goldenticket (*Tmem173^gt^*) mice, which carry a missense point mutation in the STING encoding gene *Tmem173*, were used. This missense mutation leads to a complete absence of STING protein production in the whole body.

Herein, we demonstrate that, in a toxic liver injury model such as tunicamycin, the absence of STING is protective conferring a detrimental role to STING, while in both dietary models, the absence of STING did not influence the overall outcome of liver injury or NAFLD progression. In fact, the only difference we demonstrate between WT and STING-deficient mice is a significant increase in insulin resistance in STING-deficient mice fed FPC. 

## 2. Materials and Methods

### 2.1. Animals

For TM studies, 8–11 wk old male wild-type control mice (C57BL/6 or XBP1^fl/fl^ on a C57BL/6 background) and STING-deficient C57BL/6J-Sting1^gt^/J mice (*Tmem137^gt^*, Jackson Laboratory, Bar Harbor, ME, USA) were used. All mice received one intraperitoneal (i.p.) injection of either TM (2 mg/Kg) in a DMSO/saline mixture or the equivalent DMSO/saline dose alone [12]. The mice were then sacrificed either 48 or 72 h post-injection. For the high fructose dietary studies, 7 wk old male C57BL/6 mice were purchased from the Jackson Laboratory (Bar Harbor, Maine) and allowed to acclimatize for 1 wk; *Tmem137^gt^* were bred in-house and used at 8 wk of age. The animals (n = 4–5 mice/group) were placed on a 60% fructose diet (Fructose, TD.89247, Envigo) for 4 wk. Finally, for the NAFLD studies, another set of C57BL/6 and *Tmem173^gt^* mice were randomly assigned (n = 4–6 mice/group) to either control chow (Chow, #5053, PicoLab, Forth Worth, TX, USA) or a Fructose–Palmitate–Cholesterol diet (FPC, TD.160785, Envigo, Indianapolis, IN, USA) for 1–16 wk. The mice on the FPC diet also received drinking water containing a 55:45 glucose-to-fructose mixture. Prior to euthanasia, the mice were fasted for 4 h. All mouse experiments were conducted in accordance with the guidelines set by the American Veterinary Medical Association. All mouse studies were reviewed and approved by the Committee on Animal Research at the University of California, San Francisco.

### 2.2. Histology and Immunohistochemistry

Formalin-fixed liver sections were stained with hematoxylin and eosin. Liver inflammation was quantified by immunohistochemistry for the pan-macrophage marker CD68, and fibrosis was assessed by staining with Sirius Red. Images were taken with a Leica DM6 B upright microscope and LAS X software (Leica Microsystems, Wetzlar, Germany), and percent staining was assessed using Image J software (Bethesda, MD, USA).

### 2.3. Gene Expression

RNA was isolated from whole liver using TRIzol (Invitrogen, Carlsbad, CA, USA) and purified with a Direct-zol RNA miniprep kit (ZymoResearch, Irvine, CA, USA). cDNA was synthesized using an iScript kit (Bio-Rad, Hercules, CA, USA), and gene expression was measured with TaqMan probe sets (Life Technologies, Carlsbad, CA, USA). TaqMan sets used were as follows: *Ifnb* (Mm00439552_s1), *Irf3* (Mm00516784_m1), *Tbk1* (Mm00451150_m1), *Ifit1* (Mm00515153_m1), *Ifit3* (Mm01704846_s1), and *Gusb* (Mm01197698_m1).

### 2.4. Serum Testing

Alanine aminotransferase (ALT) and total cholesterol were measured in mouse serum using an ADVIA 1800 autoanalyzer (Siemens Healthcare Diagnostics, Deerfield, IL, USA) in the clinical chemistry laboratory at the Zuckerberg San Francisco General Hospital.

### 2.5. Glucose Tolerance Testing

At 15 wk of experimental FPC feeding, a glucose tolerance test was performed. Mice were fasted for 4 h prior to testing. Blood glucose was measured at time 0 followed by a glucose bolus of 20 mg/kg i.p. Subsequent glucose levels were monitored at 15, 30, 60, and 120 min. 

### 2.6. Lipid Quantitation

Lipids were extracted from frozen liver tissue using the Folch method [13]. Total liver triglycerides were measured spectrophotometrically as previously described (TR0100, Millipore-Sigma, Burlington, MA, USA) [14].

### 2.7. Statistics

All results are reported as mean ± SEM. Statistical analysis was performed using Prism 9.1.1 software (GraphPad Software, San Diego, CA, USA). When two groups were compared, a Student’s *t*-test was applied. When more than two groups were compared, a one-way analysis of variance (ANOVA) was employed followed by a Tukey’s multiple comparisons test. *p* values <0.05 were considered statistically significant. 

## 3. Results

**STING deficiency protects against short-term toxic liver injury induced by tunicamycin.** Whole liver protein was analyzed in order to confirm the absence of STING in *Tmem173^gt^* mice (Figure 1A). The animals were injected once with TM or vehicle control (DMSO in saline) and were euthanized at either 48 or 72 h later. TM-induced liver damage was assessed by ALT levels (Figure 1B). At 48 h post-TM, both the wild-type (WT) and *Tmem173^gt^* mice demonstrated similar ALT levels. However, at 72 h post-TM, *Tmem173^gt^* mice demonstrated significantly lower ALT levels than their WT counterparts. All DMSO-injected mice demonstrated normal ALT levels. The data demonstrate a significant role for STING in developing liver damage following short-term toxic injury. However, the data also demonstrate that the role of STING is time-dependent and only significant at 72 h post-TM delivery.

**Absence of STING during short-term high fructose feeding does not affect liver injury outcome.** Knowing that STING played a significant role in toxic-induced liver damage, we wanted to investigate whether STING was also involved in other models of liver injury, such as NAFLD and high fructose feeding. WT and *Tmem173^gt^* mice were fed the high-fructose diet for 4 wk. Histologically, both groups demonstrated the same morphology, with limited steatosis and very limited damage noted (Figure 2A). The main histological findings were a few inflammatory foci per whole liver section as well as a rare mitotic figure or dead cell. ALT levels confirmed the histological findings and showed only a slight elevation for both groups after 4 wk of fructose feeding. While *Tmem173^gt^* mice tended to have higher ALT levels, the differences were not statistically significant (Figure 2B). We also investigated downstream STING targets, *Tbk1*, *Irf3*, and *Ifnb* by gene expression and found no difference between the two groups (Figure 2C). Together, these data demonstrate that short-term high fructose feeding and the possible resultant ER stress [15] may not be sufficient to trigger STING activation and its downstream signaling cascade. Furthermore, the absence of STING did not make a difference in overall liver outcome and cell health.

**The absence of STING does not protect mice from steatosis or liver injury in response to FPC feeding.** Given our mixed results regarding the contribution of STING to TM liver injury and high fructose feeding, we decided to investigate the role of STING in a more aggressive NASH-inducing model and thus fed mice a Fructose–Palmitate–Cholesterol (FPC) diet for varying time points. Both WT and *Tmem173^gt^* mice were fed either control chow or FPC for 1, 4, or 16 wk. Chow-fed mice from both genotypes demonstrated no histological abnormalities throughout the experimental time-points (Figure 3A). FPC-fed mice began to exhibit hepatic steatosis as early as 1 wk, but there was no apparent difference between WT and *Tmem173^gt^* mice (Figure 3A). By 4 wk of FPC feeding, significant steatosis could be seen in both WT and *Tmem173^gt^* FPC-fed mice, but no significant liver injury was noted, as underscored by the lack of elevation in ALT levels (Figure 3A,B). At 16 wk, both WT and *Tmem173^gt^* FPC groups demonstrated significant liver steatosis and injury, clearly demonstrating a NASH phenotype, as underlined by their liver TG, ALT values, and histological features (Figure 3A–C). Of note, the *Tmem173^gt^* FPC-fed mice appeared to develop both micro- and macrosteatosis, compared to their WT counterparts, which displayed mostly macrosteatosis. However, throughout the varying time points, no significant differences in overall steatosis or injury were noted between the WT FPC-fed mice and their *Tmem173^gt^* counterparts.

**STING neither protects nor worsens liver fibrosis or inflammation in the dietary NASH model FPC.** In order to fully understand the potential role of STING during NASH and long-term liver injury, we examined liver inflammation and fibrosis in our 16 wk animals. Liver sections were stained with Sirius Red and positive staining quantified for all groups (Figure 4A,B). Both FPC-fed mice accumulated significantly more collagen deposition by 16 wk than their chow counterparts; however, no significant difference was seen between WT and *Tmem173^gt^* mice. This was further confirmed by gene expression analysis for *Col1a1* (Figure 4C). The same was true for liver inflammation. Both FPC-fed groups had significantly more CD68+ cells than their chow counterparts, but no significant difference was noted between WT and *Tmem173^gt^* mice. These findings were confirmed by gene expression levels for *Ccl2* and *Tnfa*, which also demonstrated significant increases in the FPC-fed mice compared to their chow counterparts but no differences between genotypes. Interestingly, the chow-fed *Tmem173^gt^* mice did have significantly more CD68+ cells than their WT counterparts.

**Insulin resistance is worse in the absence of STING during long-term FPC feeding.** We further investigated the WT and *Tmem173^gt^* mice for abnormalities in their metabolic function following FPC feeding (Figure 5). FPC-fed WT and *Tmem173^gt^* mice showed significant increases in their serum cholesterol compared to their chow counterparts at all timepoints; however, no significant differences between the two genotypes were seen (Figure 5A). A glucose tolerance test was administered and showed that while both FPC-fed groups had significantly higher blood glucose levels in the beginning, only the *Tmem173^gt^* FPC mice remained high for the duration of the test (Figure 5B,C). Taken together, these data point towards a heightened state of insulin resistance for the *Tmem173^gt^* FPC mice compared to all other groups at 16 wk.

**Whole body STING deficiency in a dietary NASH model has conflicting effects on the type I IFN pathway.** Finally, we investigated whether FPC feeding activates STING signaling in the liver and whether these signals were suppressed in STING-deficient mice by qtPCR. Little to no changes were noted prior to 16 wk in both *Tbk1* and *Irf3* expression levels, but by 16 wk, a paradoxical significant decrease in expression levels was noted in both FPC-fed groups (Figure 6). Again no significant difference was noted between genotypes. The expression patterns for both *Ifit1* and *Ifit3* were unique. Both genes demonstrated a significant decrease in *Tmem173^gt^* FPC-fed mice at the 4 wk timepoint compared to WT chow mice, with non-significant trends of being lower than their WT counterparts. At 16 wk, the *Ifit1* expression levels for *Tmem173^gt^* FPC-fed mice were significantly lower than all other groups. While at 16 wk, the *Ifit3* levels for *Tmem173^gt^* FPC-fed mice were significantly lower than their wild-type counterparts. *Ifnb*, the final molecule in the STING pathway demonstrated no significant differences whatsoever, neither by timepoint or by diet, nor by genotype group. Taken together, this points to a limited role for STING in NASH pathogenesis.

## 4. Discussion

In the present study, we demonstrated a stimulus-dependent role for STING in liver injury. In TM-treated mice, the absence of whole-body STING appeared to be beneficial to liver health as indicated by reduced ALT levels 3 d post-injection. However, in contrast, STING appeared to play no role in mice-fed high fructose diet for 4 wk, although admittedly the overall injury due to high fructose feeding was very limited. These data thus indicate that the role of STING in toxic liver injury may in fact be stimulus-, time-, and intensity-dependent. Furthermore, when mice were fed a FPC NASH-inducing diet for up to 16 wk, the absence of whole-body STING appeared to play a very limited role in overall health, affecting only insulin resistance but not liver health, for example, having no effect on levels of steatosis, inflammation, or fibrosis. Based on previously published reports using HFD and MCD models, these results were unexpected [9,10,11,16]. We propose a role for lipotoxcity in the activation of STING, possibly through ER stress activation during HFD, that may not be as prominent in a diet higher in carbohydrate. 

Other groups have shown an important role for STING in both HFD and MCD models. One group, for example, using a HFD feeding model for 12 wk demonstrated that *Tmem173^gt^* mice developed significantly lower ALT, liver triglyceride content, and inflammation [11]. Another group using a MCD feeding model established that *Tmem173^gt^* mice had significantly less steatosis, ballooning, inflammation, and fibrosis at 8 wk [10]. However, in contrast, another group also using a HFD model demonstrated a beneficial role for IRF3, a downstream molecule of STING [9]. This study established that mice, in the absence of IRF3, when fed HFD developed significantly more liver injury and fibrosis, as demonstrated by a significant increase in ALT and Sirius red staining, respectively [9]. This study thus indicates a potential beneficial role for STING. In our present study, however, we demonstrated no difference in steatosis, injury, or fibrosis in the absence of STING. One significant difference between our current study and those mentioned above is the type of diet given to the animals. All of these other studies used high fat diets, whereas we used a diet high in both fat and carbohydrate; it could be that in the presence of high carbohydrates the role of STING is less important. It could also be that our diet is overall less inflammatory than HFD and thus does not interact with the STING pathway as aggressively as in these other studies. Additionally, the FPC diet was originally used in describing the role of TAZ (Transcriptional Co-Activator With PDZ-Binding Motif) in hepatic fibrogenesis, and it could be that this pathway outweighs the potential role of STING during this dietary insult [17].

An interesting finding was a significant increase in CD68% cells in *Tmem173^gt^* chow-fed mice compared to WT chow-fed mice. It has been previously documented that STING is highly expressed in macrophages [10,11] within the liver. Recently, a group demonstrated that HFD fed *Tmem173^gt^* mice with wild-type bone marrow had a significant increase in their hepatic macrophage populations, as shown by increased F4/80 staining [11] compared to *Tmem173^gt^* mice with *Tmem173^gt^* bone marrow. This group also demonstrated a significant decrease in overall hepatic macrophages when *Tmem173^gt^* mice were fed MCD diet. While we appear to show the opposite findings in our STING mutant mice, both findings beg the question whether STING may be involved in macrophage migration into the liver. 

One unexpected finding was the decrease in *Irf3* gene expression levels in our *Tmem173^gt^* mice compared to WT mice. However, this has in fact already been described by two other groups [9,18]. Furthermore *Ifnb* gene levels were not affected in either of the models employed in this paper, but no other papers report on *Ifnb* gene expression levels during a NAFLD stimulus, while one group does demonstrate a significant increase in IFNβ protein levels with HFD [19]. It may be that these dietary models are not inflammatory enough to induce *Ifnb* with STING recruitment. 

Overall, while our study demonstrates a detrimental role for STING during chemically induced toxic liver injury, it did not demonstrate a role for STING in a dietary models of NAFLD. Furthermore, we demonstrated no important role for STING in liver steatosis, inflammation, and fibrogenesis during short- and long-term feeding of a NASH-inducing diet, although we did note a significant reduction in whole-body insulin resistance in the presence of STING.

## 5. Conclusions

In conclusion, while we cannot currently pinpoint why in our studies STING appeared to only play a role in chemically induced liver injury and not dietary-induced injury; we hypothesize that it is likely due to the dietary models we chose. All previous studies were conducted with high fat-type diets, while we used diets high in carbohydrate, specifically fructose. It may be that fructose has a different effect, or no effect at all, on STING activation. The exact role of STING in sterile liver injury, specifically NASH appears to be stimulus-dependent, with different dietary models inducing different degrees of STING involvement.

## Figures and Tables

**Figure 1 nutrients-14-04029-f001:**
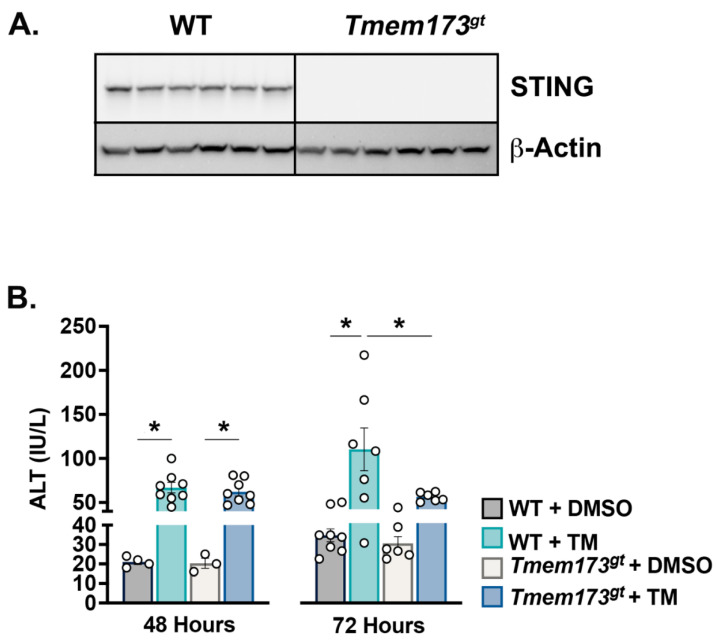
**Tunicamycin-induced liver injury is limited 3 days post-injection in the absence of STING.** (**A**) Western blot of whole liver samples from WT and *Tmem173^gt^* mice probed with a STING antibody. (**B**) ALT levels either 48 h or 72 h post-TM injection in both WT and *Tmem173^gt^* mice. Graphs represent means ± SEM. * *p* < 0.05 between highlighted groups.

**Figure 2 nutrients-14-04029-f002:**
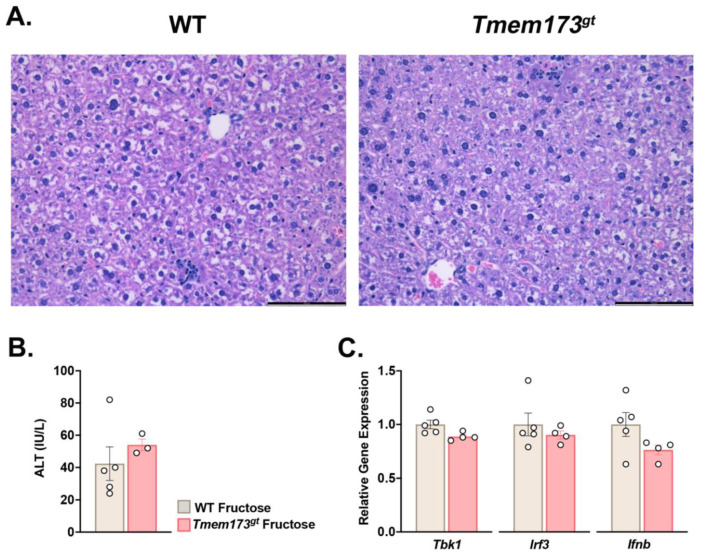
**Liver histology in WT and STING-deficient mice after 4 wk of high-fructose feeding.** (**A**) Representative photomicrographs of H&E-stained livers of high-fructose fed WT and *Tmem173^gt^* mice. Chow livers, not shown, showed no abnormalities. Bar size = 100 μm. (**B**) ALT levels of WT and *Tmem173^gt^* mice fed a high-fructose diet for 4 wk. (**C**). Relative gene expression levels for *Tbk1*, *Irf3*, and *Ifnb* for WT and *Tmem173^gt^* mice. Graphs represent means ± SEM.

**Figure 3 nutrients-14-04029-f003:**
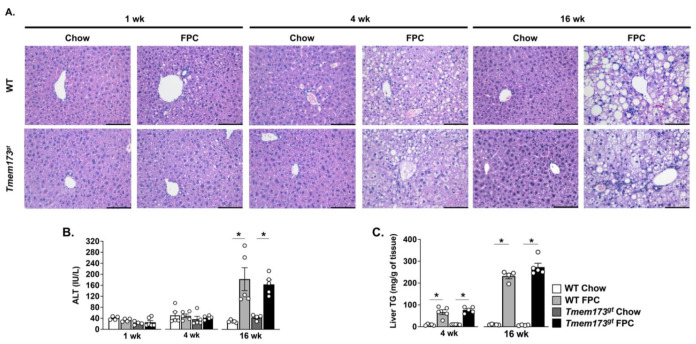
**Liver steatosis and injury in WT and STING-deficient livers during short- and long-term chow and FPC feeding.** (**A**) Representative photomicrographs of H&E stained liver from WT and *Tmem173^gt^* mice at 1 wk, 4 wk, and 16 wk post-FPC feeding. Control chow histology not shown, showed no pathological differences. (**B**) ALT levels at 1 wk, 4 wk, and 16 wk and (**C**) liver triglyceride levels for 4 wk and 16 wk in WT and *Tmem173^gt^* mice fed chow or FPC. Graphs represent means ± SEM. * *p* < 0.05 for each genotype compared to its respective chow control.

**Figure 4 nutrients-14-04029-f004:**
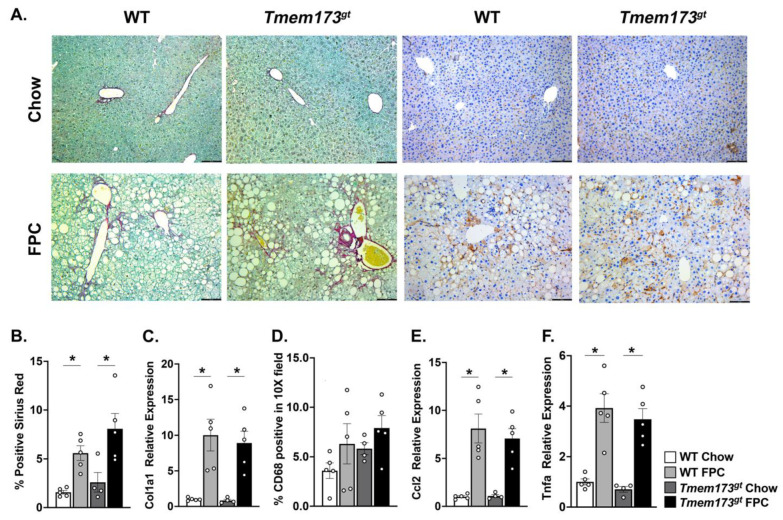
**Liver fibrosis and inflammation in WT and STING-deficient livers after 16 wk of chow or FPC feeding.** (**A**) Representative photomicrographs for Sirius red (left) and CD68 (right) for WT and *Tmem173^gt^* mice at 16 wk. Bar size = 100 μm (**B**) Quantitation of Sirius red positive staining in WT and *Tmem173^gt^* mice after 16 wk of chow or FPC feeding. (**C**) Relative gene expression levels for *Col1a1* at 16wk feeding. (**D**) Quantitation of percent of CD68 positive staining at 16 wk post chow or FPC feeding in WT and *Tmem173^gt^* mice. (**E**,**F**) Relative gene expression levels for *Ccl2* and *Tnfa* at 16 wk post-chow or FPC feeding in WT and *Tmem173^gt^* mice. Graphs represent means ± SEM. * *p* < 0.05 compared to respective chow.

**Figure 5 nutrients-14-04029-f005:**
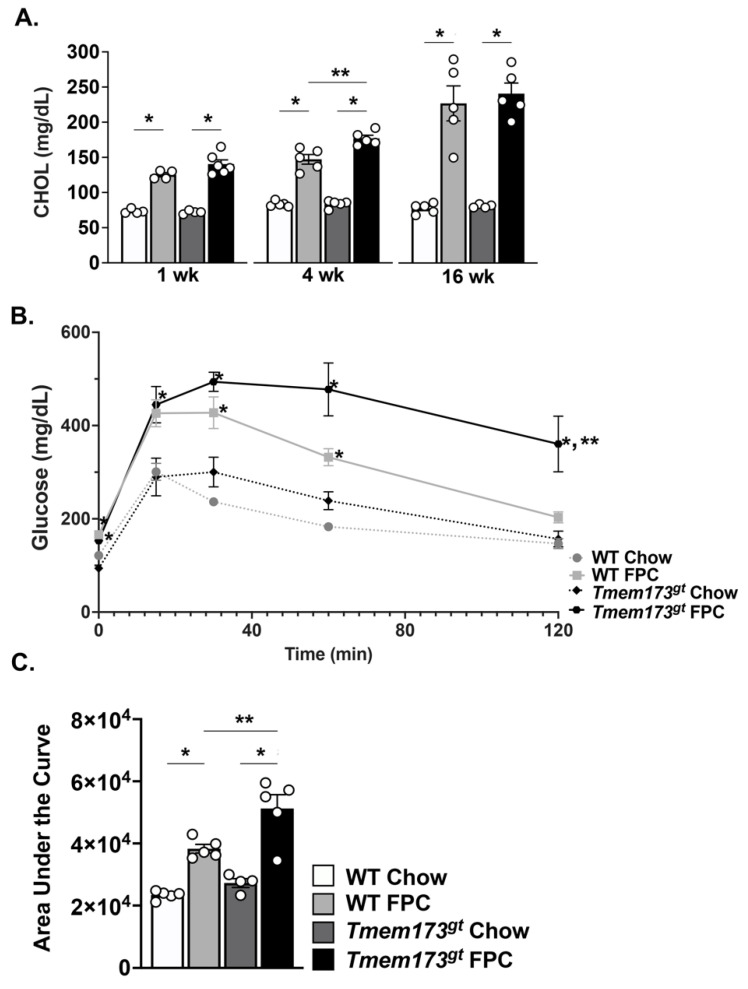
**Metabolic measurements in WT and STING-deficient mice during short- and long-term chow and FPC feeding.** (**A**) Serum cholesterol levels were measured at 1 wk, 4 wk, and 16 wk post- chow or FPC feeding in WT and *Tmem173^gt^* mice. (**B**) Glucose tolerance test was performed at 15 wk. (**C**) Area under the curve calculated from the glucose tolerance curves above. Graphs represents means ± SEM. * *p* < 0.05 compared to chow counterpart. ** *p* < 0.05 compared to WT FPC mice.

**Figure 6 nutrients-14-04029-f006:**
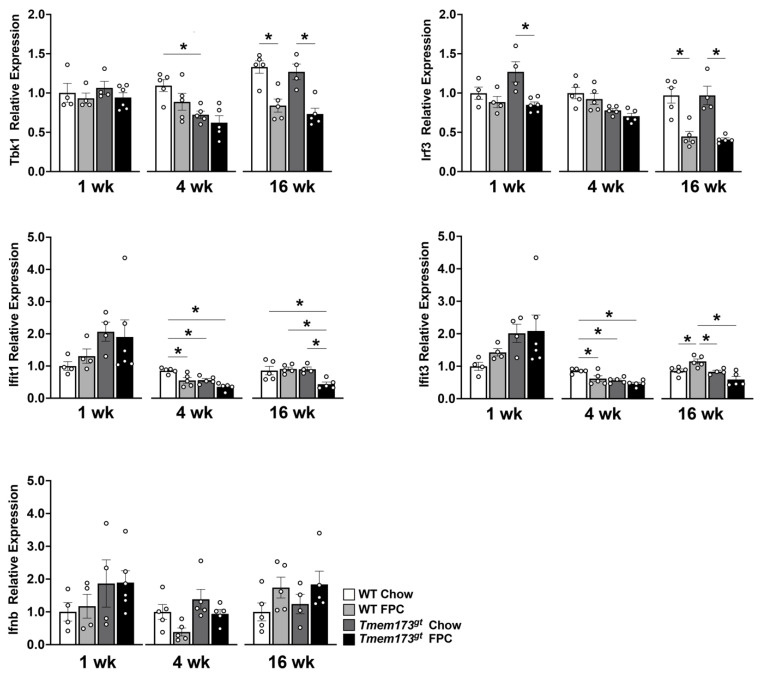
**Gene expression levels of STING downstream pathways.** Relative expression of *Tbk1*, *Irf3*, *Ifit1*, *Ifit3*, and *Ifnb* at 1, 4, and 16 wk following chow or FPC feeding in WT and *Tmem173^gt^* mice. Graphs represents means ± SEM. * *p* < 0.05 to respective group.

## Data Availability

Not applicable.

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
