# Peer review of "The Role of STING in Liver Injury Is Both Stimulus- and Time-Dependent"

_nutrients, 2022, doi:10.3390/nu14194029_

Round 1
Reviewer 1 Report
Siao et al., explored the role of STING in liver injury using different models. There are some merits. However, the manuscript needs to be revised before further consideration. My comments are listed below:
Since the objective of your study is to compare the role of STING in liver injury under different models, you should clarify what is the difference between each model in the introduction, which helps readers to understand why you treated mice with tunicamycin and FPC. Even tough you briefly mentioned about it in the discussion, more discussions are needed to emphasize the differences of each model and and potential mechanism behind each model. And why does high carbohydrate diet cause different response in liver compared to high fat diet.
Obviously, tunicamycin causes acute endoplasmic reticulum stress in liver which is different from chronic stress caused by high fat or high carbohydrate diet. This should be emphasized as well. Is it possible that the level of ER stress influences the activation of STING in liver? It will be useful to compare some ER stress markers in different models to further help understand whether ER stress may play a role in regulating STING during liver pathogenesis.
Line 12 add "of" behind "the exact role"
Line 26-37 references should be provided.
weeks or wk? please be consistent through the whole manuscript.
Pay attention to the space between dot and the next sentence. You ALWAYS leave more space between each sentence. For example, Line 10, line 11 and line 12. Please check the whole manuscript. Besides, leave space between the last word of each sentence and your reference. For example, Line 45 CC4[4,7] should be changed to CC4 [4,7].
Line 48 give full name of MCD before use you its abbreviation.
Line 68-69 how did you choose the dose? References should be provided.
Line 142 add "to be" after "tend"
Line 143 insert "," between "higher" and "the difference"
Line 145 insert "," between "Together" and "these data"
Line 146 did you test ER stress markers?
Line 148 Based on what leads you to make the conclusion that the absence of STING did not affect cell death? I did not see any data related to cell data if I did not miss something.
Line 162,170, 185, 207 and 280 add "," after "however"
Line 168-170 it seems that WT mice had bigger lipid droplets than Tmem1173tg mice in the liver under FCP diet at 16wk.
Line 228 gene name should be italic
Line 232 add "," after "Take together"
Line 243 please change to "was limited, indicating..."
Line 245 change "an" to "a"
Line 246 change "and" to "but"
Line 267 give the full name of TAZ
Line 270 change "bene" to "been"
Line 279 gene name should be italic
Line 291 delete extra dot.
Author Response
Overall we would like to thank the reviewer for a very thorough review and feel this has greatly improved our manuscript.
Since the objective of your study is to compare the role of STING in liver injury under different models, you should clarify what is the difference between each model in the introduction, which helps readers to understand why you treated mice with tunicamycin and FPC. Even tough you briefly mentioned about it in the discussion, more discussions are needed to emphasize the differences of each model and potential mechanism behind each model. And why does high carbohydrate diet cause different response in liver compared to high fat diet.
We thank the reviewer for this comment and have added more information into the introduction and discussion to address these points.
Obviously, tunicamycin causes acute endoplasmic reticulum stress in liver which is different from chronic stress caused by high fat or high carbohydrate diet. This should be emphasized as well. Is it possible that the level of ER stress influences the activation of STING in liver? It will be useful to compare some ER stress markers in different models to further help understand whether ER stress may play a role in regulating STING during liver pathogenesis.
We appreciate the reviewer’s comment and have added a small discussion on this point. However, we have not included ER stress markers as they are not influenced by FPC feeding and thus do not feel their inclusion would add to the current body of work. This is an area of further research however.
Line 12 add "of" behind "the exact role"
The authors thank the reviewer for this suggestion, and have added “of” to the sentence (now line 11)
Line 26-37 references should be provided.
References have been added, thank you for catching this omission (reference added on line 28).
weeks or wk? please be consistent through the whole manuscript.
This has been standardized throughout the manuscript as “wk”.
Pay attention to the space between dot and the next sentence. You ALWAYS leave more space between each sentence. For example, Line 10, line 11 and line 12. Please check the whole manuscript.
We appreciate this comment, however, the authors prefer maintaining 2 spaces after each period, a stylistic preference. We have checked the entire manuscript to make sure this is standardized throughout.
Besides, leave space between the last word of each sentence and your reference. For example, Line 45 CC4[4,7] should be changed to CC4 [4,7].
Thank you for this comment, we have edited the style to include a space between reference and previous word.
Line 48 give full name of MCD before use you its abbreviation.
Thank you for catching this omission, MCD has now been defined (now line 46).
Line 68-69 how did you choose the dose? References should be provided.
We have added a reference for the dose chosen, thank you.
Line 142 add "to be" after "tend"
Thank you for comment, this has been edited and now reads “While Tmem173gt mice did tend to have higher ALT levels, the differences were not statistically significant (Figure 2B). ” (now line 140)
Line 143 insert "," between "higher" and "the difference"
Thank you for finding this omission, this has been edited.
Line 145 insert "," between "Together" and "these data"
Thank you for finding this omission, this has been edited.
Line 146 did you test ER stress markers?
We thank the reviewer for their comment. We did not measure ER stress markers in this model but have done so in our previous paper where we fed mice high fructose diet for 4wk. The levels are very low, and thus we felt it would not add to the paper and have omitted.
Line 148 Based on what leads you to make the conclusion that the absence of STING did not affect cell death? I did not see any data related to cell data if I did not miss something.
We appreciate the reviewer’s comments. We based our discussion of cell death on ALT levels as they should increase in the presence of hepatocellular death.
Line 162,170, 185, 207 and 280 add "," after "however"
Thank you for finding these omissions, this has been edited.
Line 168-170 it seems that WT mice had bigger lipid droplets than Tmem1173tg mice in the liver under FCP diet at 16wk.
We appreciate the reviewer’s comment and have edited the sentence to clarify our point (now lines 167-171)
Line 228 gene name should be italic
Thank you for finding this omission, this has been edited.
Line 232 add "," after "Take together"
Thank you for finding this omission, this has been edited.
Line 243 please change to "was limited, indicating..."
This has now been edited to read “These data thus indicate”
Line 245 change "an" to "a"
Thank you for finding this error, this has been edited.
Line 246 change "and" to "but"
Thank you for finding this omission, this has been edited.
Line 267 give the full name of TAZ
Thank you for catching this omission, TAZ has now been defined.
Line 270 change "bene" to "been"
Thank you for finding this error, this has been edited.
Line 279 gene name should be italic
Thank you for finding this error, this has been edited.
Line 291 delete extra dot.
Thank you for finding this error, this has been edited.
Reviewer 2 Report
The topic is interesting, although the models used are far away from human NASH. I have some suggestions.
Minor points:
1. Please include tunicam in the title and add respective data in the abstract.
2. Discuss this paper: Jiang JJ, Zhang GF, Zheng JY, Sun JH, Ding SB. Targeting mitochondrial ROS-mediated ferroptosis by quercetin allivates high-fat diet-induced hepatic lipotoxicity. Front Pharmacol 2022; 13: 876550.Are your models also based on ferroptosis concept?
3. Abstract needs complete rewording, difficult to read. Abbreviations must be expanded when first mentioned, add actual numbers of the most important parameters with means and SD/SEM and p value.
4. Legend of Figure 1 needs more data , means? Asterics stand for what?
Author Response
- Please include tunicam in the title and add respective data in the abstract.
We thank the reviewer for this comment and have edited the abstract accordingly. For the title we have simplified it to now say: “The role of STING in liver injury is both stimulus- and time-dependent.”.
- Discuss this paper: Jiang JJ, Zhang GF, Zheng JY, Sun JH, Ding SB. Targeting mitochondrial ROS-mediated ferroptosis by quercetin allivates high-fat diet-induced hepatic lipotoxicity. Front Pharmacol 2022; 13: 876550.Are your models also based on ferroptosis concept?
We thank the reviewer for highlighting this article, however, we do feel ferroptosis is not within the scope of this work and thus have not investigated it enough to include the above reference in this body of work. We do not believe ferroptosis plays a critical role in these models.
- Abstract needs complete rewording, difficult to read. Abbreviations must be expanded when first mentioned, add actual numbers of the most important parameters with means and SD/SEM and p value.
We thank the reviewer for this comment and have reworked the abstract, as well as added small amounts of data.
- Legend of Figure 1 needs more data , means? Asterics stand for what?
We thank the reviewer for finding this omission and have corrected it.
Round 2
Reviewer 1 Report
The manuscript has been improved.
Line 13 the format of "with" is different from the other text. Please revise.
Conclusion
Please include the response is time-dependent in the conclusion.